# Increase Paclitaxel Sensitivity to Better Suppress Serous Epithelial Ovarian Cancer via Ablating Androgen Receptor/Aryl Hydrocarbon Receptor-ABCG2 Axis

**DOI:** 10.3390/cancers11040463

**Published:** 2019-04-02

**Authors:** Wei-Min Chung, Yen-Ping Ho, Wei-Chun Chang, Yuan-Chang Dai, Lumin Chen, Yao-Ching Hung, Wen-Lung Ma

**Affiliations:** 1Graduate Institution of Clinical Medical Science, and Graduate Institute of BioMedical Sciences, School of Medicine, China Medical University, Taichung 40403, Taiwan; qqrice68@yahoo.com.tw (W.-M.C.); wei66@iris.seed.net.tw (W.-C.C.); 2Sex Hormone Research Center, Department of Obstetrics and Gynecology, and Reproductive Medicine Center, China Medical University Hospital, Taichung 40403, Taiwan; d25939@mail.cmu.edu.tw (Y.-P.H.); silviachen6585@gmail.com (L.C.); 3Department of Pathology, Ditmanson Medical Foundation Chia-Yi Christian Hospital, Chia-Yi City 60002, Taiwan; cych03884@gmail.com; 4Department of OBs & GYN, BenQ Medical Center, Suzhou 215004, China; 5Department of Nursing, Asia University, Taichung 41354, Taiwan

**Keywords:** ovarian cancer serous subtype, androgen receptor, ABCG2, aryl hydrocarbon receptor, paclitaxel resistance

## Abstract

Background: Epithelial ovarian cancer (EOC) is one of the most lethal gynecological malignancies and presents chemoresistance after chemotherapy treatment. Androgen receptor (AR) has been known to participate in proliferation. Yet the mechanisms of the resistance of this drug and its linkage to the AR remains unclear. Methods: To elucidate AR-related paclitaxel sensitivity, co-IP, luciferase reporter assay and ChIP assay were performed to identify that AR direct-regulated ABCG2 expression under paclitaxel treatment. IHC staining by AR antibody presented higher AR expression in serous-type patients than other types. AR degradation enhancer (ASC-J9) was used to examine paclitaxel-associated and paclitaxel-resistant cytotoxicity in vitro and in vivo. Results: We found AR/aryl hydrocarbon receptor (AhR)-mediates ABCG2 expression and leads to a change in paclitaxel cytotoxicity/sensitivity in EOC serous subtype cell lines. Molecular mechanism study showed that paclitaxel activated AR transactivity and bound to alternative ARE in the ABCG2 proximal promoter region. To identify AR as a potential therapeutic target, the ASC-J9 was used to re-sensitize paclitaxel-resistant EOC tumors upon paclitaxel treatment in vitro and in vivo. Conclusion: The results demonstrated that activation of AR transactivity beyond the androgen-associated biological effect. This novel AR mechanism explains that degradation of AR is the most effective therapeutic strategy for treating AR-positive EOC serous subtype.

## 1. Introduction

Epithelial ovarian cancer (EOC) is among the most lethal gynecological malignancies and has a variety of cellular origins, histological characteristics, and therapeutic responses [1]. In terms of histology, EOCs can be divided into four subtypes that account for different percentages of the overall population of EOC patients, namely the serous (~75%), mucinous (~10%), endometrioid (~10%), and clear-cell (~3%) subtypes [2,3]. A significant proportion of EOC patients are diagnosed at the advanced stage, and these patients have a poor five-year survival rate (<30%) after receiving surgery or chemotherapy [4,5]. Among such patients, those who are most numerous and have the poorest prognosis are those with serous EOC [6,7]. The first-line chemotherapies for treating advanced-stage EOC are the paclitaxel/carboplatin combination and the cisplatin/carboplatin combination, and these treatments have a response rate of around 80% [8,9]. Unfortunately, however, approximately 70–80% of serous EOC patients subsequently develop chemoresistance and relapse [4], with this high rate of relapse being considered a major concern among medical providers. 

ATP-binding cassette transporters (ABC transporters) are a type of transmembrane protein pumping drugs out of cells [10] by consuming the energy released through ATP hydrolysis [11]. The breast cancer resistance protein/ATP-binding cassette subfamily G member 2 (BCRP or ABCG2) belongs to the ABC transporter superfamily, which has been reported in ovarian cancer stem cells and is associated with chemoresistance [12,13]. A recent study showed that the ABCG2 transporter might be potential target for preventing paclitaxel therapy failure [14]. However, all of those inhibitors failed in clinical trials due to complications or ineffectiveness [13,14]. As such, the targeting of ABCG2 to prevent chemotherapy resistance is still controversial in the field. 

Androgen, a type of steroid hormone, has been suggested to potentially play a role in ovarian pathophysiology [15,16]. Androgen acts by binding to the androgen receptor (AR), a transcription factor belonging to the nuclear receptor superfamily [17,18]. The activated AR translocates into the nucleus and binds to androgen response elements (AREs) with co-regulators to regulate gene expression (this process is also called “transactivation”) [17,18,19]. A number of studies have shown that androgen/AR signaling plays various roles in promoting the growth and/or progression of EOC [20,21]. Furthermore, association studies using immunohistochemical approaches have revealed discrepant expressions of AR in different EOC subtypes [21,22]. Expression of AR and its functions in EOC have been studied for a long period of time; however, there is still much controversy [21,23,24,25]. Therefore, the pathophysiological role of AR in EOC requires a detailed characterization. 

In this study, we discovered a preference in AR expression in serous subtype EOC which causes the promotion of cancer growth. Specifically, we found that treating serous EOC cells with paclitaxel, the standard-of-care chemotherapy agent, surprisingly causes the activation of an AR–AhR complex →ABCG2 regulatory axis that in turn causes EOC cells to become resistant to paclitaxel.

## 2. Results

### 2.1. AR/AhR Co-Regulate ABCG2 Expression in Serous EOC

Given that previous studies have shown paclitaxel sensitivity influenced by ABCG2 expression and function [26,27], we examined AR expression in three serous EOC cell lines (HeyA8, OVCAR3, and SKOV3ip1) (Appendix A). We then manipulated AR expression via adding AR-cDNA or AR-shRNA in these cells (Appendix A) to test their relation to ABCG2 expressions. The results demonstrated that AR-cDNA robustly enhanced ABCG2 protein expression (Figure 1A). In contrast, the knockdown of AR expression diminished ABCG2 mRNA expression (Figure 1B) in EOC cells. Using the Hoechst expulsion assay to evaluate ABCG2 function with flow cytometry [28], we found that ABCG2 efflux ability was positively regulated by AR expression in EOC cells (Figure 1C). These results indicated that AR is highly associated with ABCG2 expression, with its effects in that regard possibly taking place at the transcriptional level. 

To identify AR-regulated ABCG2 expression, we surveyed a promoter of the ABCG2 genome (+336/−3628 kb) from the National Center for Biotechnology information (NCBI) database (accession no. NC_018915) (Appendix A). According to previous studies [29,30] and our recent study in AhR [31], we hypothesized that the AR–AhR complex could also be observed in EOC cells. As demonstrated in Figure 1D, using a co-immunoprecipitation (co-IP) assay we were able to observe AR–AhR interactions in the three serous EOC cell lines. These results showed that AR and AhR were physically interacting. 

To characterize the possibility that the AR–AhR complex mediates ABCG2 expression, the ABCG2 promoter sequence (Appendix A) was subjected to prediction using the PROMO website Available online: http://alggen.lsi.upc.es/cgi-bin/promo_v3/promo/promoinit.cgi?dirDB=TF_8.3 (Accessed on 2 April 2019) to find AREs and DREs. There were several putative AREs (−2642/−3110) and a core DRE (−194/−190) that were found (Figure 1E, Appendix A). Interestingly, there were two alternative AREs (AAAGAAT) located on the proximal region of the 5′-promoter [32]. Those alternative AREs existed on −546/−553 and −959/−966 of the ABCG2 promoter (Figure 1E, Appendix A). In order to determine the transactivation activity of the alternative AREs and the core DRE, various promoter fragments of ABCG2 were constructed, as shown in Figure 1F. 

We constructed fragments of +336 ~ −523 (4F4R; contains the core DRE), +336 ~ −1336 (2F4R; contains both alternative AREs and the core DRE), and –3268 ~ −1898 (1F1R; contains putative AREs) on respective pGL3-basic luciferase plasmid (Figure 1F). We then transiently transfected those plasmids in EOC cells and treated the cells with either DHT or TCDD to measure luciferase activities. To identify the core DRE transactivity, the 4F4R-pGL3 was tested for transcriptional function. The result showed that none of the treatments activated 4F4R-pGL3 transactivity (Appendix A, upper panel); however, TCDD could successfully activate 4×DRE-TATA-Luc [33] luciferase activity, where 4×DRE-TATA-Luc played the role of positive control (Appendix A, down panel). This result indicated the relevant fact that a core DRE could not be directly turned on by AhR to promote ABCG2 expression in EOC serous cells. To further examine the effects of AR on ABCG2 promoters, 1F1R-pGL3 and 2F4R-pGL3 were tested for transactivation function. We found that 1F1R-pGL3 presented an absence of luciferase activity under all treatments (Appendix A). However, TCDD turned on transactivation activity in the 2F4R-pGL3 construction, whereas the DHT treatment did not induce transactivation activity in it (Figure 1G). However, we failed to observe the androgen/AR transactivation function while constructing those AREs on reported constructs (Figure 1G and Appendix A). These results indicated that TCDD induced AR/AhR translocation activity that interacted with the alternative AREs within the proximal ABCG2 promoter region.

### 2.2. AR Reduces Sensitivity and Develops Resistance to Paclitaxel in Serous EOC

To study the potential impact of AR on paclitaxel therapy to suppress the EOC progression, we assayed their impact on the colony formation capacity. The results showed that increasing AR expression by adding AR-cDNA reduced paclitaxel-related colony forming inhibition (Figure 2A) and cytotoxicity (Figure 2B). In contrast, decreasing AR expression via adding AR-shRNA increased paclitaxel-related colony forming inhibition (Figure 2A) and cytotoxicity (Figure 2C). Cytotoxicity IC_50_ of each set of manipulated-cells is shown in Appendix A. 

We then tested the effects of AR in the development of paclitaxel resistance. We observed the cell growth dynamics under a low-dose, long-term treatment with paclitaxel (0.05 µM, 7 days) (Figure 2D, AR-cDNA; Figure 2E, AR-shAR). The results showed that the EOC cells were sustained under paclitaxel treatment when AR was expressed, whereas the cells were vulnerable when AR expression was knocked down. These data suggest that AR may play a role to modulate the paclitaxel resistance.

To evaluate AR/ABCG2 expression in patients with different EOC subtypes, we performed a single cohort study with immunohistochemistry (IHC) using AR antibody. The review found that the tissue samples consisted of 16 serous, 20 clear-cell, 29 mucinous, and 49 endometrioid EOC subtype samples, and AR IHC was then performed on these samples (Figure 3A, and quantitation in Figure 3B). We found that AR expression was predominantly abundant in the serous subtype samples (Figure 3Aa,b), while AR expression was very low in the clear-cell (Figure 3Ac,d), mucinous (Figure 3Ae,f), and endometrioid (Figure 3Ag,h) subtype samples. To confirm our in vitro findings that presented in patients, serial sections of AR vs ABCG2 IHC slides subjected to a quantitation and association approach showed that AR and ABCG2 expressions were highly correlated (Figure 3C,D; R^2^ = 0.4641).

We further used the web-based gene survivor Kaplan–Meier Plotter (KM plotter; http://kmplot.com/analysis/index.php?p=background) to evaluate AR mRNA expression in relation to disease prognosis. We restricted our analysis to stage 3 and 4 serous EOC samples, we found AR expression to be a positive prognostic marker (hazard ratio (HR) = 1.48; *p* = 0.01; Figure 3E). Moreover, when stage 3 and 4 serous EOC samples were subjected to optimum debulk surgery followed by chemotherapy (taxel and platin), the impact of AR expression on the overall survival (OS) was found to be even more significant (HR = 1.66; *p* = 0.0015; Figure 3F).

Overall, these data indicate that AR expression is biased in serous EOC, and that its pathological role could be related to chemoresistance/chemosensitivity. 

### 2.3. Paclitaxel Induces AR/AhR–ABCG2 Axis in Serous EOC

To dissect the molecular mechanism of AR/AhR–ABCG2 regulation, we questioned whether TCDD acts as an exogenous ligand in activating AhR. Although AR/AhR interaction showed a potential regulatory mechanism of ABCG2 expression, it was hard to explain the association between TCDD and paclitaxel-treated EOC patients. Thus, we proposed the alternative possibility that paclitaxel acts as an alternative inducer to activate AR/AhR, which, in turn, regulates ABCG2 expression. To evaluate this possibility, the expression of ABCG2 mRNA was analyzed under paclitaxel treatment in the three serous cell lines. The data showed that paclitaxel treatment resulted in higher expression of ABCG2 in the treated cells compared to the control cells (Figure 4A). Encouraged by this finding, we further investigated whether paclitaxel induced AR/AhR transactivity by using the 1F1R-pGL3 and 2F4R-pGL3 constructions. The data indicated that 1F1R-pGL3 showed an absence of transactivity (Appendix A), whereas 2F4R-pGL3 presented significantly increased luciferase activity (Figure 4B) under paclitaxel treatment. These findings indicate that AR/AhR-regulated ABCG2 expression is significantly induced by paclitaxel. 

To further verify the role of AR/AhR transactivity in regulating ABCG2 expression, a chromatin-immunoprecipitation (ChIP) assay [34] was used to examine AR occupancy of alternative AREs in the proximal ABCG2 promoter region. To perform the ChIP assay, pairs of primer were designed to target the three ARE fragments in the ABCG2 promoter region. These DNA fragments were ChIP1 (−2569/−2688), ChIP2 (−799/−1004), and ChIP3 (−500/−608) (Figure 4C). Using AR antibody, the ChIP assay showed that AR was recruited to interact with the promoter region of ChIP2 and ChIP3 (which contained the alternative ARE of 5′-AAAGAAT) under paclitaxel treatment in the three EOC serous cell lines. In addition, ChIP1 (which contained the putative ARE of 5′-CAAATGTCC, 5′-GGACATGAA) was not available for AR binding (Figure 4D). These data indicate that AR directly enhances ABCG2 expression through binding to alternative ARE in the presence of paclitaxel.

### 2.4. Degrading AR for Paclitaxel Insensitivity/Resistance Therapy in Serous EOC 

As the paclitaxel → AR/AhR → ABCG2 regulatory axis had been elucidated, we then wanted to test the value of targeting AR for EOC therapy. There are two compounds available for interrupting this axis by interfering with AR function. One is the new anti-androgen enzalutamide (or MDV-3100), which was developed for the irreversible blockade of androgen–AR interactions, and which has already been proven to offer a survival benefit to relapsed prostate cancer patients [35]. The other is ASC-J9, a newly developed AR degradation enhancer, which has been proven successful in pre-clinical studies [36,37]. 

Thus far, the blocking of androgen/AR function with MDV3100 (enzalutamide) in EOC has been proposed [38]; however, such blocking remains hypothetical. In order to compare the effects on cytotoxic capacity of the anti-androgen vs AR degradation enhancer, MDV3100 and ASC-J9 were combined, respectively, with paclitaxel, and then compared in terms of the resulting cytotoxic effects. The results showed that the combination of paclitaxel and ASC-J9 had a greater synergistic effect than that of paclitaxel combined with MDV3100 (Figure 5A). Cytotoxicity IC_50_ of the combination therapy for each set of cells is shown in Appendix A. Since ASC-J9 exhibited excellent therapeutic potential in EOC cells when combined with paclitaxel, we then wanted to further dissect its clinical value. We therefore treated the three EOC serous cell lines with paclitaxel, ASC-J9, and paclitaxel combined with ASC-J9, respectively, and the cell viability assay showed that 5 µM of ASC-J9 and 5 nM of paclitaxel exert little effect in terms of cytotoxicity. However, the paclitaxel combined with ASC-J9 exhibited a synergistic cytotoxic effect (Figure 5B). This significant synergistic effect of ASC-J9 and paclitaxel was presented in a dose-dependent manner with further additions of paclitaxel (Figure 5C). Using a colony formation assay to observe the long-term effects on the EOC cells, we found that paclitaxel combined with ASC-J9 could abolish the colony formation capacity compared to other treatments (Figure 5D). 

In addition to observing the effects of ASC-J9 in terms of solving paclitaxel insensitivity, we also wanted to examine its effects on paclitaxel resistance. We thus established paclitaxel-resistant (PTXR) OVCAR3 cells using an add-on dosing and long-term treatment procedure [26,27]. The PTXR cells exhibited paclitaxel resistance compared to naïve OVCAR3 cells (Figure 6A). In the PTXR cells, we found that the expressions of AR, AhR, and ABCG2 were robustly upregulated compared to the parental cells (Figure 6B). Therefore, the data indicate that the paclitaxel → AR/AhR → ABCG2 axis might form a positive feedback regulatory loop for cancer to develop chemoresistance. We found that ASC-J9 combined with paclitaxel therapy was more effective than paclitaxel alone in these PTXR cells (Figure 6C). By combining a low dose of ASC-J9 with the paclitaxel treatments, the cytotoxic effect of paclitaxel was significantly improved (Figure 6D). In order to test the ASC-J9/paclitaxel synergistic effect in vivo, mice with subcutaneous tumors bearing PTXR cells were treated with low-dose paclitaxel (10 mg/kg), ASC-J9 (40 mg/kg), or the combination. We treated the mice when the tumor size reached 400 mm^3^, injecting the given intraperitoneal treatment two times/week for four consecutive weeks. The tumor sizes for the mice treated with vehicle vs paclitaxel vs ASC-J9 alone remained comparable. However, a significant reduction in tumor size was seen in the mice receiving the paclitaxel with ASC-J9 co-treatment (*p* < 0.05) (Figure 6E). After the mice were sacrificed, the tumors were weighed and photographed (Figure 6F). 

Taken together, the data presented in Figure 6 suggest that targeting AR in paclitaxel-resistant serous EOC could be a new therapeutic strategy. A summary of our understanding of the paclitaxel → AR/AhR → ABCG2 regulatory axis mechanisms in serous EOC cells is presented in Figure 7.

## 3. Discussion

Serous EOC is the subtype of EOC that affects the most women patients, has the poorest prognosis, and is the quickest to develop chemoresistance [1,9,39]. In this study, we found that paclitaxel could be the activator of the AR/AhR → ABCG2 regulatory axis, through which it would induce chemotherapy resistance and further promote serous EOC progression. Targeting AR with ASC-J9 is thus a potential therapeutic strategy for treating the serous subtype of ovarian cancer. Figure 7 consists of a diagram of the positive feedback loop of paclitaxel → AR/AhR → ABCG2 for chemotherapy resistance. The new mechanism of chemoresistance discovered in this report is discussed further below.

### 3.1. New Mode of AR Activation by Exposure to Paclitaxel

Sun et al. showed that AR and ABCG2 were upregulated under paclitaxel treatment and in paclitaxel-resistant EOC cells [26,27]. These studies implied that AR is potentially associated with ABCG2 expression and chemoresistance. In analyzing ABCG2 promoter sequences, we did not observe either classical AREs or AR transactivation of the putative ARE-constructed pGL3-luciferase activity. This was consistent with the findings of previous studies [30,40], and furthermore implied an indirect or alternative activation by androgen/AR of ABCG2 gene expressions. However, Tan et al. [29], for example, reported that AhR is a transcriptional activator of ABCG2. Meanwhile, an alternative ARE was reported by Wu. et al. [32]; however, whether an environmental stimulant is involved in AR transactivation remains unknown. Our own previous work showed that the interaction of AR/AhR promoted endometrial cancer progression under AhR ligand (TCDD) treatment [31]. In the present report, we found that there is an alternative ARE, located on the ABCG2 promoter, which could be activated by the xenobiotic receptor AhR, and form an AR/AhR complex. This AR/AhR complex could be activated by TCDD, as well as by paclitaxel (Figure 4). 

Except for ABCG, there is another ABC transporter gene family related to chemotherapy resistance, the ABCBs [11]. Among them, ABCB1 had been reported associated with paclitaxel resistance in ovarian cancer [41,42,43]. For example, Januchowski et al. established multiple drug- resistant clones from A2780 OVCA cells to be resistant to cisplatin, paclitaxel, and doxorubicin [42], and the authors demonstrated that ABCB1 expression was upregulated in paclitaxel-resistant A2780 cells. In another study published by the same group, they presented consistent results in a primary culture of ovarian cancer cells and concluded that ABCB1 plays a critical role in paclitaxel resistance. In addition, Sun et al. determined ABCB1 gene expression and regulation in paclitaxel-resistant ovarian cancer (SKOV3) [43]. They found ABCB1 expression was much higher in resistant cells than parental cells. In that study, Sun et al. found both DHT and paclitaxel activated AR transactivity and regulated ABCB1 gene expression [42,43]. To further evaluate the direct promotion of ABCB1 expression by AR, the ABCB1 promoter region was analyzed [44]. They identified that several classical AREs existed in the promoter region and showed direct regulation of AR for ABCB1 expression [44]. In addition to ABCB1, Sun et al. stated that the high expression of ABCG2 in resistance cells might contribute to taxel resistance. In the current study, we have provided direct proof of the AR/AhR–ABCG2 axis contributing to paclitaxel resistance.

In previous studies, Bailey et al. analyzed the ABCG2 promoter region and data showed none of the classical AREs existed in the promoter [30]. Consistently, Scotto KW reviewed the transcription factors of ABC transporters [45]. This report presented all potential transcription factors for every ABC transporter, including ABCB1 and ABCG2. However, AR was not mentioned in the transcription factor analysis of all ABC transporters [45]. According to accumulated studies, the important role of AR in regulating ABCB1 expression has been shown [26,27,44]. In our study, we performed different AR transactivation functions in regulating ABCG2 expression. Even though ABCG2 plays a minor role in paclitaxel resistance, our study has shown novel transactivation actions in ABCG2 expression, including paclitaxel-induced transactivity, alternative ARE existing in the ABCG2 promoter region, and the recruitment of AhR in the AR-ABCG2 axis. In this study, we found AR/AhR–ABCG2 expression under paclitaxel treatment. This also implicates potential regulation of AR/AhR on ABC transporters. This discovery explained the regulatory process of paclitaxel resistance at the molecular level. 

### 3.2. Novel Therapeutic Strategy by Degrading AR in EOC Tumors 

The roles of androgen/AR signaling in ovarian cancer progression have been reported previously [20], and some therapeutic strategies targeting androgen/AR have been tested either in experimental models or clinical settings [38,46]. However, the therapeutic efficacy of the androgen/AR blocking reagents used in those tests were not significant, or were inconclusive in regards to ovarian cancer therapy. Recently, the US FDA has approved olaparib, a poly(ADP-ribose) polymerase inhibitor (PARPi), as a novel therapy for treating ovarian cancer [47,48]. The results of a single-arm phase II study showed that patients with germline breast related cancer antigen 1/2 (BRCA 1/2)-mutated advanced ovarian cancer who had received three or more prior lines of chemotherapy had a median progression-free survival (PFS) of 7 months, a median overall survival (OS) of 16.6 months, and a 1-year OS rate of 64.4% [47]. Furthermore, the studies of olaparib used in women with high-grade serous ovarian cancer (HGSOC) recurrence reported that patients who received olaparib maintenance treatment showed a longer PFS (median, 8.4 vs. 4.8 months; HR, 0.35; 95% confidence interval (CI), 0.25–0.49; *p* < 0.001; data maturity, 58%) [49,50]. The clinical trials of PARPi showed encouraging therapeutic results in treating EOC patients. However, patients receiving PARPi treatment are restricted by following conditions: (1) germline BRCA1/2 mutation (it has been reported that approximately 30% of HGSOC presents BRCA1/2 mutation and epigenetic silencing) [48,51], (2) PARPi is not used in first-line chemotherapy reagents, and (3) PARPi is a drug resistance considered to be related to ABCG2 transporters [52,53]. In our findings, we have indicated that AR degradation enhancer ASC-J9 provided a synergistic effect with the first-line chemotherapeutic agent paclitaxel in serous EOC (Figure 5 and Figure 6). Thus, in addition to advocating for further testing of the combination of paclitaxel and AR, we suggest that targeting AR combined with PARPi may remedy the deficiencies of PARPi treatment and provide an advanced therapeutic strategy for serous EOC.

## 4. Materials and Methods 

### 4.1. Patient Tissue Study

The paraformaldehyde-embedded EOC tissue samples analyzed in this study were obtained from patients diagnosed with EOC from 2008 to 2013 at the China Medical University Hospital (CMUH, Taichung, Taiwan), and clinical data for these patients was acquired from the Cancer Registry Database of CMUH. The subtypes of EOC patients were selected from ward charts. Patient tissue slides were confirmed by two boarded pathologists to exclude ambiguity or mixing histology between subtypes. A total of 114 tissue samples were collected (16 serous subtype, 20 clear-cell subtype, 29 mucinous subtype, and 49 endometrioid subtype samples). Access to the tissue samples was approved by the Internal Review Board of the China Medical University Hospital (#CMUH104-REC2-075). 

### 4.2. Immunohistochemistry and Quantitation of Staining Score

The histological studies were performed with modifications as described in previous studies [31,54]. For histologic inspection, we treated tissue sections (2 μM) with hematoxylin and eosin, or stained sections with antibody specific to AR and ABCG2 while using an ABC kit (Vector Laboratories, CA, USA) to enhance the staining signals. The staining distributions for AR and ABCG2 were graded using a five-point scale according to the percentage of positive staining (0: <1%; 1: 1–20%; 2: 20–40%; 3: 40–60%; 4: 60–80%; 5: 80–100%). The staining scores were summed and showed a correlation between the distributions for AR and ABCG2 expression. The slides were independently examined by Dr. Yuan-Chang Dai (Pathology Dept., Ditmanson Medical Foundation Chia-Yi Christian Hospital, Chia-Yi City, Taiwan), who was blind to the clinicopathological data.

### 4.3. Cell Lines and Reagents 

The human EOC serous subtype cell lines (HeyA8, OVCAR3, and SKOV3ip1) and the human embryonic kidney cell line (HEK293T) used in this study were grown in Dulbecco’s Modified Eagle medium (DMEM) (GIBCO, Carlsbad, CA, USA) with 10% fetal bovine serum (FBS) (GIBCO) and 1% penicillin/streptomycin (GIBCO) at 37 °C in a humidified atmosphere of 5% CO_2_. The cells were then cultured in fresh DMEM/10% CD-FBS (charcoal-dextran stripped FBS) for 24 h before the experiments. The ovarian cancer cell lines were provided courtesy of Dr. Min-Chie Huang (MD Anderson, TX, USA), and the HEK293T cells were obtained from Dr. Yuh-Pyng Shyr (Center of Molecular Medicine, China Medical University Hospital, Taichung, Taiwan). The chemotherapeutic drugs used in this study included paclitaxel, MDV3100 (enzalutamide) (Sigma-Aldrich, St. Louis, MO, USA), and ASC-J9, which was kindly provided by Dr. Chawnshang Chang (University of Rochester Medical Center, Rochester, NY, USA). 

### 4.4. Immunoblotting Assay

Cells were first washed with 1× PBS and resolved in RIPA buffer (100 mM Tris, 5 mM EDTA, 5% NP40; pH 8.0) with protease inhibitors (1 mM phenyl-methyl sulphonyl fluoride, 1 μg/mL aprotinin, 1 μg/mL leupeptin). Proteins were resolved by SDS-PAGE and then transferred to PVDF membranes. The blocking of non-specific binding was accomplished by adding 5% non-fat milk. After the application of primary antibody (AR (N-20), Santa Cruz, CA, USA; β-actin, Santa Cruz, CA, USA; AhR (H-211), Santa Cruz, CA, USA; BCRP/ABCG2 (BXP-21), Abcam, Cambridge, UK), secondary antibody (1:3000, HRP-goat-anti-mouse and HRP-goat-anti-rabbit) were applied for 1 h at room temperature. Signals were enhanced using an ECL chemiluminescence kit (Millipore, Danvers, MA, USA) and detected by ChemiDoc XRS+ (BioRad, Hercules, CA, USA).

### 4.5. Transfection and Lentivirus Infection Procedure 

Cells were transfected with the following lentivirus plasmids: psPAX2 packaging plasmid, pMD2G envelope plasmid (Addgen, Watertown, MA, USA), pLKO.1-shLuc (luciferase shRNA), pLKO.1-siAR (National RNAi Core Facility, Academia Sinica, Taipei, Taiwan), pWPI-vector ctrl, and pWPI-hAR (Addgen). Lentiviral plasmids were co-transfected with psPAX2 and pMD2G into HEK293T cells at a ratio of 3:2:4 by lipofectamine 2000 (Invitrogen, Carlsbad, CA, USA) according to the manufacturer’s instructions. After 6 h, the medium was replaced with fresh DMEM/10% FBS medium, and the cells were maintained at 37 °C in a humidified incubator in an atmosphere of 5% CO_2_ for 48 h. The medium containing the virus was collected by centrifugation and filtered through a 0.45 μm filter. Medium containing 0.8 mg/mL polybrene (Sigma-Aldrich) was then added to culture dishes containing 10^6^ EOC cells. After 16 h of infection, the medium containing virus was replaced with fresh DMEM/10% FBS medium, and the cells were maintained at 37 °C in a humidified incubator in an atmosphere of 5% CO_2_ for 48 h. The infected cells were then collected and analyzed. 

### 4.6. Total RNA Isolation and cDNA Synthesis

RNA was extracted from EOC cells. Briefly, cells that had reached 80%~90% confluence in 100-mm dishes were lysed with 1 mL Trizol (Invitrogen). Phenol/chloroform was then added, and RNA-rich layers were separated by centrifugation. Soluble RNA was precipitated with 2-propanol. The RNA was then rinsed with 75% ethanol and dissolved in RNase-free water. For first-strand cDNA synthesis, 5 μg of total RNA was used to perform reverse transcription PCR using the PrimeScript RT reagent kit (TAKARA Bio Inc., Kyoto, Japan). cDNA was synthesized according to the manufacturer’s instructions.

### 4.7. Quantitative Real-Time PCR Analysis

A real-time detection system (Bio-Rad Laboratories, Inc., CA, USA) and the KAPA SYBR FAST One-Step qRT-PCR Kit (KAPABIOSYSTEMS, MA, USA) were used according to the manufacturers’ instructions. Relative gene expression was determined by normalizing the expression level of the target gene to the expression level of a housekeeping gene (β-actin). Threshold value (Ct) dynamics were used (2^−ΔΔCt^) for the quantitation of gene expression. The quantitative real-time PCR primer sequences are shown in Appendix A.

### 4.8. Cell Viability Assay and IC_50_ Values

The water-soluble tetrazolium salt (WST-1) reagent (Roche, Basel, Switzerland) was used to assess cell viability. Briefly, 4 × 10^3^ cells/100 μL/well were seeded in 96-well plates with DMEM/10% FBS. The cells were exposed to paclitaxel (0.01, 0.05, 0.1, 0.5, 1 μM) in culture medium at 37 °C for 48 h. Then, 10 μL of WST-1 solution was added to each well, and the cells were allowed to incubate at 37 °C in an incubator for 1 h. Cell viability was quantified by colorimetric detection in an ELISA plate reader (BECKMAN, Il, USA, COULTER PARADIGM Detection Platform) at an absorbance of 450 and 690 nm to generate an OD proportional to the relative abundance of live cells in the given wells. The readings of the measured values of 50% inhibition concentration (IC_50_) [55] for each set of cells were determined by CalcuSyn software [56] (BioSoft, Cambridge, UK)

### 4.9. Colony Formation Assay and Standard Cell Number Count 

Two sets of 1.5 × 10^5^ cells/dish were seeded in 6-cm plates with DMEM/10% CD-FBS and incubated for 14 days. In one set of cells, 1000 μL of 4% formaldehyde solution was added to fixed cells, and the cells were allowed to incubate at room temperature for 1 h. Crystal violet cell staining was then performed. After 1 h, the crystal violet was washed from the cell culture dish, and the cell colonies were photographed. The other set of cells were subjected to cell counting. 

### 4.10. ABCG2 Efflux Capacity Assay

ABCG2 efflux analysis was performed as described previously by Goodell et al. [28]. Briefly, 10^6^ cells were detached and washed, then incubated in DMEM containing 2% FBS and 5 µg/mL Hoechst33342 dye (Sigma-Aldrich) for 90 min at 37 °C, either alone or in the presence of 50 µM verapamil (Sigma-Aldrich). After certain incubation time periods, the cells were washed with ice-cold 1× PBS and then resuspended with 1× PBS supplemented with 2% FBS and 2 µg/mL propidium iodide (Sigma-Aldrich) at 4 °C for 5 min to exclude dead cells. The cells were analyzed using flow cytometry (BD, CA, USA, LSR II Flow Cytometry) with dual wavelength analysis (Hoechst-blue, 424–444 nm; Hoechst-red, 675 nm) after excitation with 350 nm UV light.

### 4.11. Genomic DNA Extraction and PCR Amplification

Genomic DNA (gDNA) was purified from EOC cells using the Quick-gDNA MiniPrep kit (ZYMO RESEARCH, Irvine, CA, USA). Extraction procedures were performed according to the manufacturer’s instructions. Before PCR amplification, gDNA was digested by the restriction enzyme BamHI and the digested gDNA fragments were used as the PCR template. Five ABCG2 promoter region fragments were amplified by PCR, namely the −3268/+362, −3268/−1396, −1396/+362, −959/+362, and −523/+362 fragments. The PCR product names and primers are shown in Appendix A. 

The conditions for the *ABCG2* promoter region PCR reactions were as follows: denaturation at 95 °C for 5 min; 45 cycles of amplification (95 °C for 30 s, 55 °C ~ 64 °C for 30 s, 68 °C for 4 min); and final extension at 68 °C for 10 min. ABCG2 promoter region fragments were completed using AccuPrim *Pfx* DNA polymerase, which allows for the high-fidelity amplification of DNA fragments for downstream plasmid construction (Invitrogen). All DNA fragments were purified and sequenced.

### 4.12. ABCG2 Promoter Region Plasmid Construction and Luciferase Reporter Assay

All the ABCG2 promoter fragments were cloned into the SacI–HindIII sites of the pGL3-basic vector plasmids. A luciferase reporter assay was performed as previously described. Briefly, pGL3-ABCG2 promoter fragments and pRL-TK (thymidine kinase promoter-driven renilla luciferase plasmid) were transiently co-transfected into cells. After 6 h, the medium was replaced with fresh medium and 10% CD-FBS. The cells were then cultured for 48 h with or without DHT (10 nM). After 24 h, the cells were washed with 1× PBS, then incubated in the presence of 100 µL cell culture lysis reagent (CCLR) (Promega, Fitchburg, WI, USA) at room temperature for 30 min. Cell lysates were then placed in a microtube and centrifuged at 14,000 rpm for 5 min. Supernatant (5 µL) was then mixed with 50 µL luciferase assay reagent. Luciferase activity was measured immediately using a luminescence microplate reader and presented as relative luminescence units.

### 4.13. Chromatin Immunoprecipitation (ChIP) 

A ChIP assay was performed as described by Mulholland et al. [34]. Cultured EOC cells were treated with DMSO and 2 nM paclitaxel overnight, then the cells were crosslinked with a final concentration of 1% formaldehyde and placed on a shaker for 10 min at room temperature. Chromatin/protein crosslinking was stopped by adding 0.125 M glycine and rocking the cells for 5 min. The growth medium was removed by washing the cells with cold PBS, then the cells were scraped into PBS in a 1.5 mL tube. Next, the cells were pelleted at 1500 rpm for 10 min at 4 °C and resuspended with 1.5 mL ice-cold cell lysis buffer (5 mM PIPES pH 8.0, 85 mM KCl, 0.5% NP40) for 10 min on ice. Nucleic acids were pelleted by centrifuging at 1000 rpm for 10 min at 4 °C. The cytosolic portion was removed and resuspended in 300 μL nuclear lysis buffer (50 mM Tris-Cl pH 8.1, 10 mM EDTA, 1% SDS, protease inhibitors) for 10 min. Chromatin was sonicated to an average of 200–500 bp (assessed by gel) by using the 6 × 20-s pulses at ~30% max. Setting on a sonicator (Qsonica Q125 sonicator with CL-18 probe, Boston Lab Co, Woburn, MA, USA) in a 15-mL conical tube. Either 2.5 μg of anti-AR antibody or an equivalent amount of nonimmune control IgG (Santa Cruz, CA, USA) was added to the lysate and inverted overnight. In order to reduce the SDS concentration to ~0.1%, lysate was centrifuged at high speed for 15 min at 4 °C. To clear non-specific binding proteins and DNA, the supernatants were pre-cleared for 2 h using 20 μL of protein A-agarose/salmon sperm DNA protein G (Millipore) beads rotated at 4 °C. The samples were then centrifuged at 3000 rpm for 2 min at 4 °C, and the supernatants were then removed from the tubes carefully. Next, 1 mL of high-salt buffer (0.01% SDS, 1% NP-40, 1.2 mM EDTA, 16.7 mM Tris-Cl pH 8.1, 167 mM NaCl) was added to all samples, which were then rotated for 10 min at room temperature. The samples were then centrifuged at 3000 rpm for 2 min at room temperature. Next, the supernatants were carefully aspirated, with this step being repeated twice. The aspirated supernatants were washed twice with TE buffer (1 mM EDTA, 10 mM Tris-Cl pH 8.0), and the samples were resuspended in 300 µL of elution buffer (50 mM Tris-Cl pH 8.0, 10 mM EDTA, 1% SDS) with 1 µl of proteinase K (20 µg/µL) and incubated overnight at 65 °C to reverse cross-links. The samples were then centrifuged at full speed for 5 min at room temperature, and the supernatants were transferred to fresh 1.5-mL tubes. The purification of the DNA fragments was performed using the PCR Clean-Up Kit (AllPure PCR Clean-Up & Purification Kit, AllBio Science, Inc, Taiwan). Elution of the DNA by 50 µL TE buffer and analysis using real-time PCR were then performed. The real-time PCR primer sequences are shown in Appendix A.

### 4.14. Co-Immunoprecipitation

EOC cells in test tubes were carefully washed twice with pre-chilled PBS. Non-denaturing lysis buffer (20 mM Tris HCl pH 8, 137 mM NaCl, 1% Nonidet P-40 (NP-40), 2 mM EDTA) was then added to the tubes. The cells were scraped off and moved to clean 1.5-ml tubes. 20 µl protein A/G agarose was added to the tubes to eliminate non-specific binding proteins, and the tubes were then shaken using a horizontal shaker for 10 min at 4 °C. The tubes were then centrifuged at 1000 g at 4 °C for 5 min, and the supernatant was transferred to new tubes while the protein A/G-agarose beads were discarded. A total of 500 μg of cell lysate per IP was used to immunoprecipitate AhR using anti-AhR antibody (Santa Cruz, CA, USA) at 4 °C, with the tubes shaken overnight. Then protein A/G plus agarose beads (Santa Cruz, CA, USA) were added directly to the supernatant and the mixture was shaken for 2 h at 4 °C. Next, it was centrifuged at 1000 g at 4 °C for 5 min, and the supernatant was removed. The beads were then washed twice with ice-cold PBS. Finally, 50 µl RIPA buffer (0.1% SDS, 150 mM NaCl, 50 mM Tris pH 8.0) and 6× sampling dye were added. The proteins were detected by western blotting using anti-AR and anti-AhR (Santa Cruz, CA, USA) antibody. Rabbit IgG was used as the control.

### 4.15. Establishment of Paclitaxel-Resistant Ovarian Cancer Cell Line

Establishment of paclitaxel-resistant OVCAR3 (PTXR) cells was performed according to Sun et al. [26,27] with modifications. In brief, the PTXR cells were obtained from parental cells by continuous treatment with increased concentrations of paclitaxel (ranging from 10–200 nM) over six months. Initially, the OVCAR3 cells were plated at 80% confluence and then treated with 10 nM paclitaxel for two months. This was followed by treatment with double concentrations of paclitaxel for another month. The paclitaxel concentrations were then doubled again for the following month. The chemoresistant cell lines were then maintained in paclitaxel-containing growth medium for further studies.

### 4.16. Animal Xenograft Transplantation

The animal handling and experimental procedures were approved by the Animal Experimental Ethics Committee of China Medical University (Ethical code: 104-100-N, 22052015). Six-weeks-old female athymic nude mice (CB17/Icr-Prkdc^scid^/Cr1Nar1) were purchased from the National Laboratory Animal Center (Taipei, Taiwan), and 5 × 10^6^ PTXR cells were subcutaneously injected into both flanks of each mouse. When the tumor volume (TV) reached 400 mm^3^, the mice were randomized into four groups (placebo, 10 mg/kg paclitaxel, 40 mg/kg ASC-J9, and combined). All the treatments were given intraperitoneally three times a week for four weeks. The tumor length and width were measured by caliper twice weekly, as was the body weight of each mouse. When the mice were sacrificed, the tumor volume was calculated as follows: TV = (width^2^ × length)/2.

### 4.17. Statistical Analysis

Statistical analyses were performed using a Student’s *t*-test. All experiments were repeated at least three times, and *p*-values < 0.05 were considered to indicate statistical significance.

## 5. Conclusions

This is the first report demonstrating that paclitaxel induces AR transactivity that directly interacts with alternative ARE in the ABCG2 promoter region to regulate gene expression. Our study also provides a perspective therapeutic insight, specifically that the degradation of AR could provide the most effective therapeutic strategy for treating AR-positive serous subtype ovarian cancers.

## Figures and Tables

**Figure 1 cancers-11-00463-f001:**
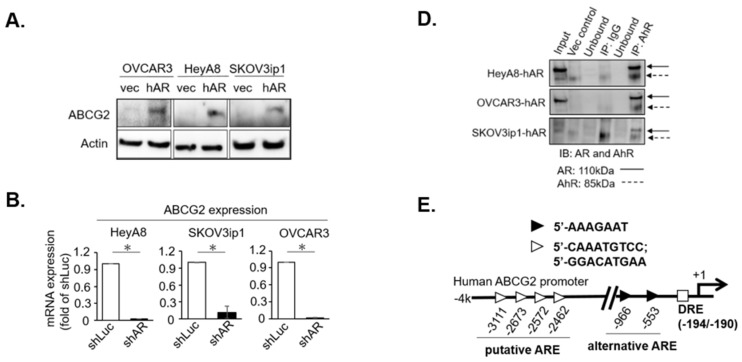
AR interacts with AhR to co-regulate ABCG2 gene expression in vitro. (**A**) Overexpression of AR enhances ABCG2 protein level. Total protein was loaded in at 40 µg/lane, and AR antibody was used to detect AR expression. Actin was used as an internal control. (**B**) Knockdown of AR attenuates ABCG2 expression in EOC serous subtype cells. RNA was extracted and cDNA was synthesized using 5 µg of total RNA. ABCG2 expression was determined using qRT-PCR. Actin was used as an internal control. Results of mRNA expression were calculated by 2^−▲▲Ct^ and normalized by shLuc. (**C**) AR enhances ABCG2 efflux capacity by detected Hoechst staining; 10^6^ cells were detached and washed, and then incubated in DMEM containing 2% FBS and 5 µg/mL Hoechst33342 dye for 90 min at 37 °C, either alone or in the presence of 50 µM verapamil. After certain incubation time periods, cells were washed with ice-cold 1× PBS and then resuspended with 1× PBS supplemented with 2% FBS and 2 µg/mL propidium iodide at 4 °C for 5 min. The cells were analyzed using flow cytometry with dual wavelength analysis (Hoechst-blue, 424–444 nm; Hoechst-red, 675 nm) after excitation with 350 nm UV light. (**D**) AR interactions with AhR are detected using co-IP. All cells were AR-overexpressed, and the protein extract of each cell line was immunoprecipitated by anti-AhR antibody. The precipitate was subjected to western blotting with the antibody against the AR and AhR. (**E**) Schematic representation of the putative AREs, alternative ARE, and core DRE (schematically indicated as ■) that exist in the ABCG2 promoter region. AREs are shown as triangles with distal (►) and proximal (►) locations as the end base from the transcriptional start site. (**F**) Various fragments of the ABCG2 promoter region are constructed on pGL3-luciferase plasmids. The arrows indicate DNA fragment locations within the ABCG2 promoter region that were amplified by PCR. Various DNA fragments that contained a putative or alternative ARE were named 1F1R or 2F4R. A DNA fragment that contained the core DRE was named 4F4R. Amplified DNA fragments constructed into pGL3-luciferase plasmids were described as 1F1R-pGL3, 2F4R-pGL3, and 4F4R-pGL3. (**G**) TCDD induced 2F4R-pGL3 luciferase activity in serous cells; 4F4R-pGL3 reporter constructs and pRL-TK internal control were co-transfected into three serous type cell lines and incubated for 24 h. After 24 h of incubation, the cells were exposed to DMSO or 10^−8^ M DHT or 5 nM TCDD for 24 h. The results indicate fold changes from the DMSO-treated, pGL3 vector-transfected control for each cell line. Each experiment was performed in triplicate (* *p* < 0.05).

**Figure 2 cancers-11-00463-f002:**
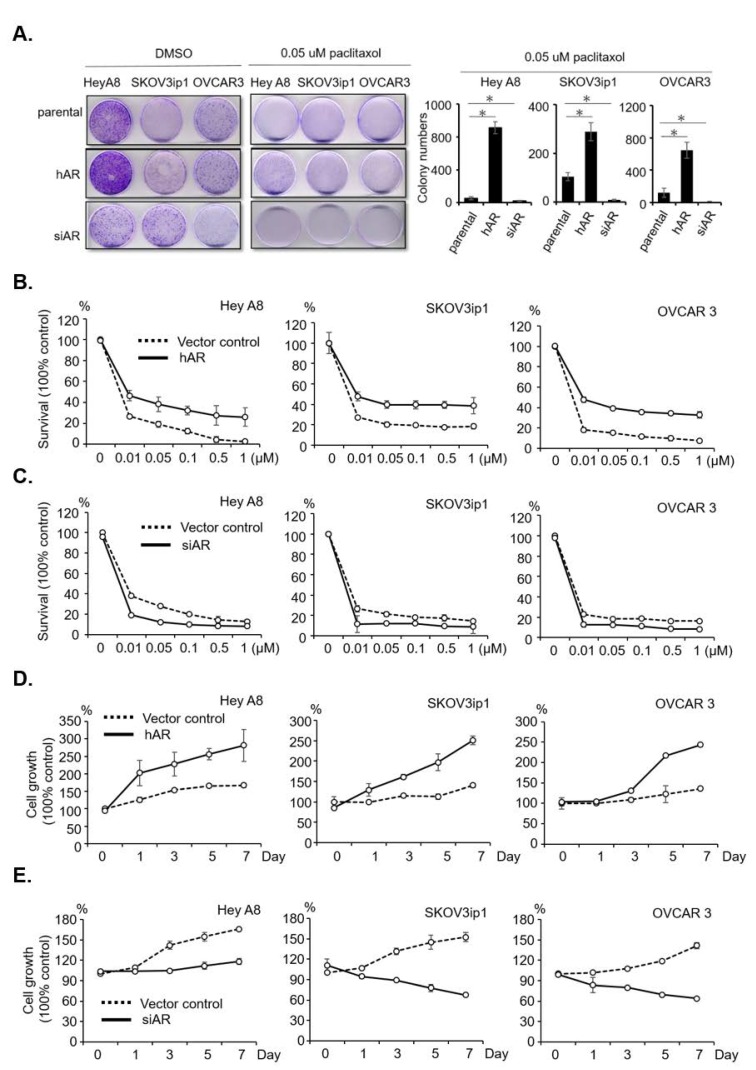
AR associates with paclitaxel sensitivity and resistance. (**A**) Differential paclitaxel colony suppression efficacy in AR-manipulated serous EOC cells. The panels on the left-hand side are images of the colonies, and the panels on the right-hand side show the quantitation of the colony numbers. (**B**) Overexpression of AR increases cell viability under paclitaxel treatment; 5 × 10^3^ cells of three serous cell lines were seeded into 96-well culture plates and treated with increased concentrations of paclitaxel (0.01, 0.05, 0.1, 0.5, 1 µM) for 48 h. The Y axis indicates absorbance at A450 nm. (**C**) Knockdown of AR decreases cell viability under paclitaxel treatment; 5 × 10^3^ cells of three serous cell lines were seeded into 96-well culture plates and treated with increased concentrations of paclitaxel (0.01, 0.05, 0.1, 0.5,1 µM) for 48 h. The Y axis indicates absorbance at A450 nm. (**D**) Overexpression of AR increases time-dependent (0, 1, 3, 5, 7 days) cell viability under 0.005 µM paclitaxel treatment; 5 × 10^3^ cells were seeded into 96-well culture plates and treated with 0.005 µM paclitaxel for 0, 1, 3, 5, 7 days. WST-1 was added at every given day. The Y axis indicates absorbance at A450 nm. (**E**) Knockdown of AR decreases time-dependent (0, 1, 3, 5, 7 days) cell viability under 0.005 µM paclitaxel treatment; 5 × 10^3^ cells were seeded into 96-well culture plates and treated with 0.005 µM paclitaxel for 0, 1, 3, 5, 7 days. WST-1 was added at every given day. The Y axis indicates absorbance at A450 nm. Each experiment was performed in triplicate (* *p* < 0.05).

**Figure 3 cancers-11-00463-f003:**
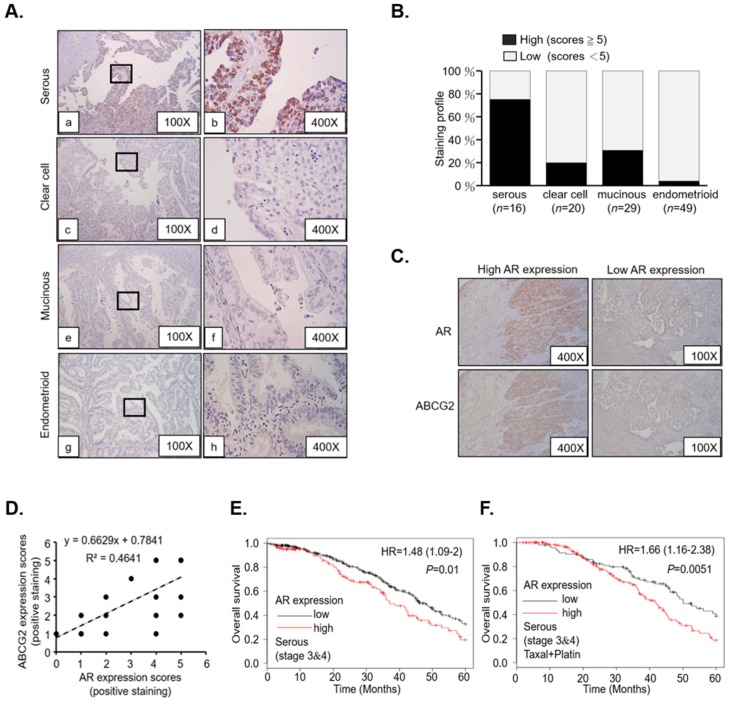
AR expressed predominantly in the serous EOC patients and is associated with ABCG2 expression and poor prognosis. All patient tissue blocks were selected by a pathologist from a China Medical University Hospital (CMUH) cohort, and IHC staining was performed using AR antibody. (**A**) IHC staining was performed using AR antibody. The expression of AR in different EOC subtypes is shown. EOC serous subtype (a,b) presented a higher expression of AR than the clear-cell subtype (c,d), mucinous subtype (e,f), and endometrioid subtype (g,h). (**B**) Demography of AR IHC staining scores related to EOC subtype patients. Two independent slide reviewers scored the slides. The scores were totaled from a reading of positive staining (1~5 point) plus a reading of intensity staining (1~5 point). A total score <5 was defined as low expression, while a total score ≥5 was defined as high expression. (**C**) IHC staining revealed AR relation to ABCG2 expression in EOCS serous subtype patient tissues. Patient tissue blocks were divided into low or high AR expression and sliced by serial dissection. IHC staining was performed using AR and ABCG2 antibody (the distribution of AR and ABCG2 were indicated by a brown color). (**D**) IHC staining scores of AR and ABCG2 show a significant positive correlation. The staining distributions for AR and ABCG2 were graded using a five-point scale according to the percentage of positive staining (0: <1%; 1: 1–20%; 2: 20–40%; 3: 40–60%; 4: 60–80%; 5: 80–100%). The staining scores were then summed and showed a distribution correlation between AR and ABCG2 expression. The statistical analysis of AR expression associated with the survival rates for the EOC subtypes was performed using the KM plotter database (http://kmplot.com/analysis/). (**E**) Serous subtype patients (at stage 3 and 4) who had a higher expression of AR were associated with a lower overall survival rate (*p* = 0.01) at five years. (**F**) Serous subtype patients (at stage 3 and 4) with a higher expression of AR and taxel and platin treatments were associated with a much lower overall survival rate (*p* = 0.0051) at five years.

**Figure 4 cancers-11-00463-f004:**
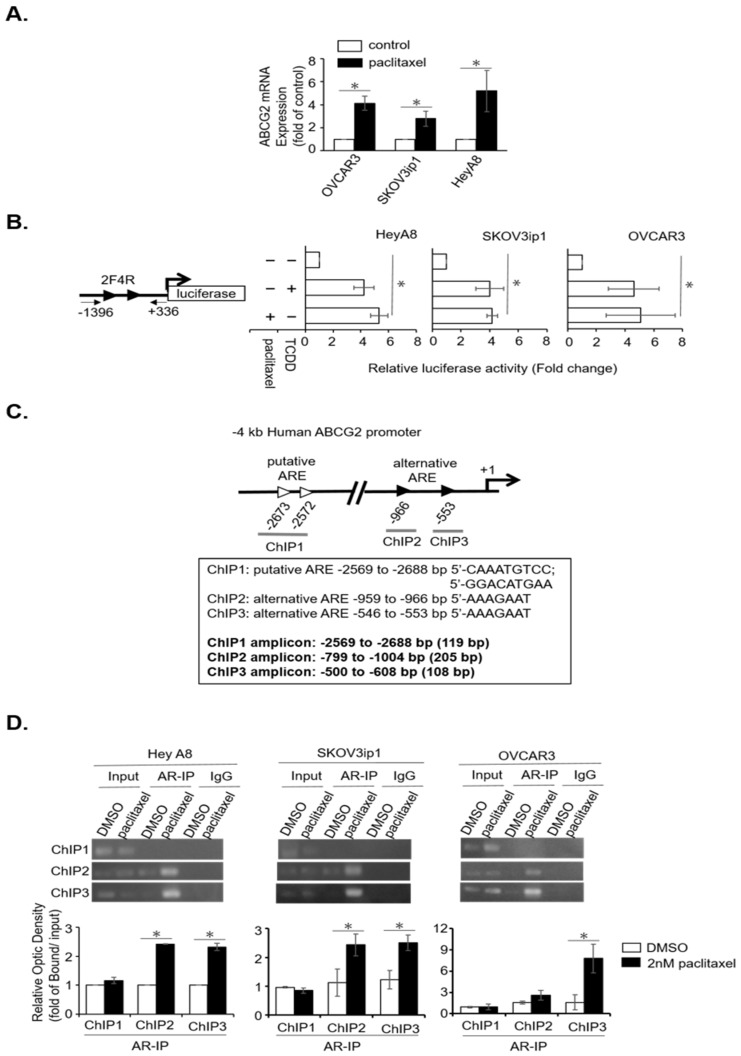
Paclitaxel-induced AR/AhR-regulated ABCG2 expression via binding to proximal alternative AREs in the ABCG2 promoter region. (**A**) ABCG2 was upregulated under paclitaxel treatment in all the serous cells. The cells were treated by 2 nM paclitaxel for 48 h and RNA was extracted and cDNA was synthesized by 5 µg total RNA. ABCG2 expression was determined using qRT-PCR. Actin was used as an internal control. Results of mRNA expression were calculated by 2^−▲▲Ct^ and normalized by DMSO control. Each experiment was performed in triplicate. (**B**) Paclitaxel promoted alternative ARE luciferase activity. The 2F4R construct (which contained the alternative ARE schematically referred to as ►) was used to test paclitaxel-induced AR transactivation. (**C**) Schematic position indicating the ChIP regions of the ABCG2 promoter (ChIP1~3). (**D**) Paclitaxel-induced AR binding on alternative ARE (-AAAGAAT-) within the proximal ABCG2 promoter. The upper panels show the representative ChIP results of PCR amplicon of ChIP1~3 in the three EOC cell lines. The lower panels show the quantitation results of ChIP experiments. Each piece of data was taken from at least three reproducible experiments, where * indicates *p* < 0.05.

**Figure 5 cancers-11-00463-f005:**
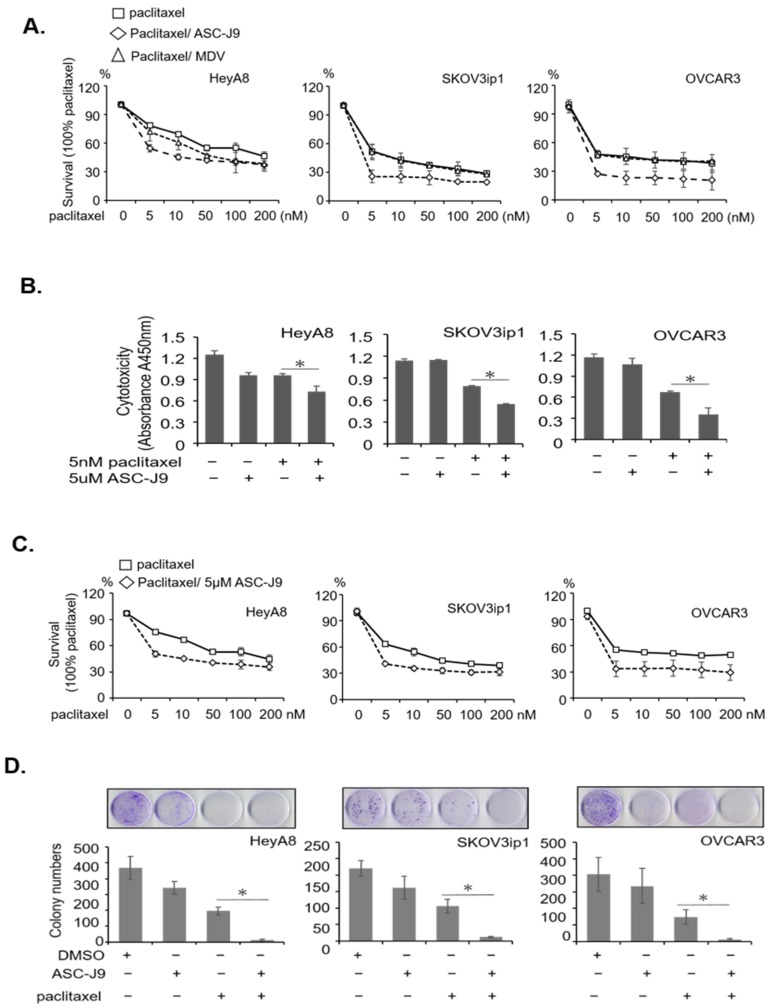
AR degradation compound ASC-J9 enhances paclitaxel therapeutic efficacy in vitro. (**A**) Paclitaxel and/or ASC-J9 effects on EOC serous cells. The cells were treated with paclitaxel (0, 5, 10, 50, 100, 200 nM) and/or 5 µM ASC-J9 or 10 µM MDV3100 to observe cytotoxicity. The Y axis indicates absorbance at A450 nm of WST-1 assay. (**B**) Cytotoxicity was measured in EOC cells. The Y axis indicates absorbance at A450 nm. (**C**) Dose-dependent cytotoxicity was measured in EOC cells. The Y axis indicates absorbance at A450 nm. Paclitaxel and/or ASC-J9 effects on EOC serous cells. (**D**) The cells were treated with paclitaxel (5 nM) and/or 5 µM ASC-J9 to test colony forming capacity. Upper panels show the colonies of the three EOC cell lines, and the lower panels show the quantitated number of colonies for each.

**Figure 6 cancers-11-00463-f006:**
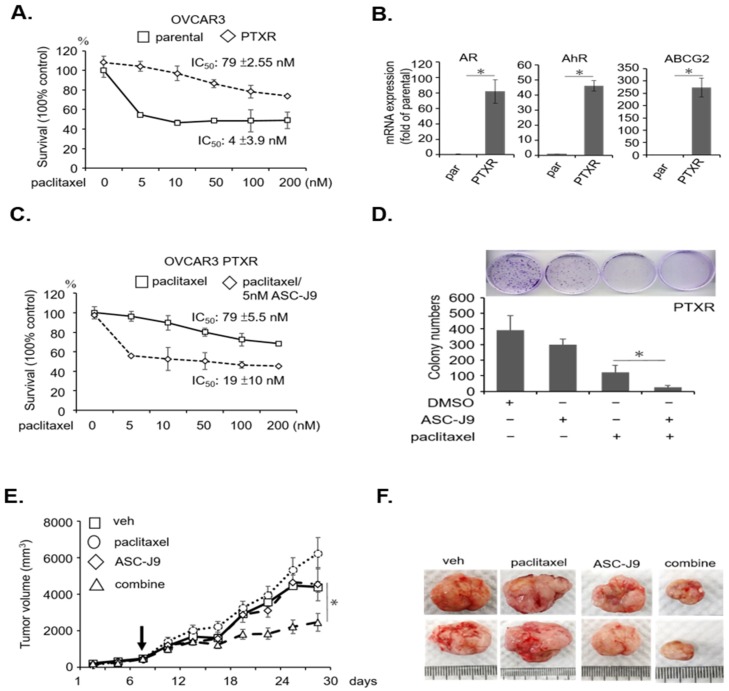
Combination therapy enhanced cytotoxicity in paclitaxel-resistant (PTXR) serous EOC cells in vitro and in vivo. (**A**) Successful establishment of OVCAR3 paclitaxel-resistant (PTXR) cells. The IC_50_ of OVCAR3 parental cells was 4 ± 3.9 nM, where PTXR was 79 ± 2.55 nM. The Y axis indicates absorbance at A450 nm. Each experiment was performed in triplicate. (**B**) Paclitaxel-resistant PTXR cells expressed higher AR, AhR, and ABCG2 mRNA compared to the parental OVCAR3 cells. (**C**) Co-treatment with ASC-J9 enhanced paclitaxel cytotoxicity in paclitaxel-resistant (PTXR) OVCAR3 cells. PTXR cell cytotoxicity was visible under either paclitaxel treatments (0, 5, 10, 50, 100, 200 nM) or combined with 5 µM ASC-J9 for 48 h. The IC_50_ of paclitaxel alone was 79 ± 5.5 nM, while that of paclitaxel and ASC-J9 was 19 ± 10 nM. (**D**) Co-treatment of ASC-J9 (5 µM) enhanced paclitaxel (10 nM)-related colony inhibition in PTXR cells. DMSO treatments served as vehicle control. (**E**,**F**) Combination treatment of paclitaxel and ASC-J9 exhibited effective therapeutic efficacy compared to single treatments. Tumor mice began the treatment procedure at the time a tumor grew to 400 mm^3^ (black arrow). The tumor suppression curves were plotted (**E**) and the representative tumors were photographed (**F**). The treatment groups were vehicle (veh; *n* = 10); paclitaxel (10 mg/kg; *n* = 10); ASC-J9 (40 mg/kg; *n* = 10); and combination (*n* = 10). All in vitro data were fromed at least three independent experiments, where * indicates a *p*-value < 0.05.

**Figure 7 cancers-11-00463-f007:**
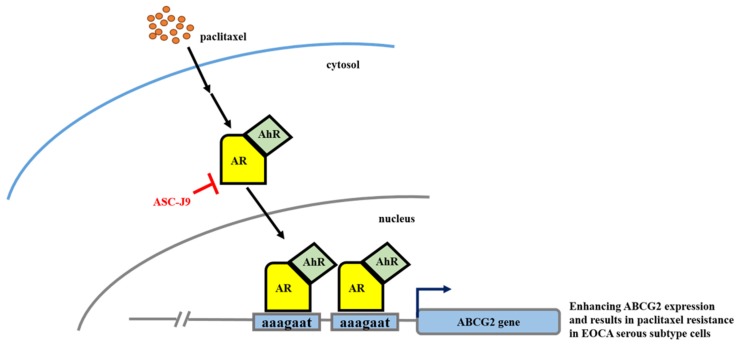
Mechanistic illustration of the paclitaxel–AR/AhR–ABCG2 regulatory axis in serous EOC chemoresistance. The treatment of paclitaxel could induce AR–AhR to form a complex and bind to the ABCG2 promoter to transactivate ABCG2 expression. The paclitaxel–AR/AhR–ABCG2 axis mediated the process of chemotherapy resistance in serous EOC cells. Interfering with AR abundance through the use of ASC-J9 could block this process.

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
