# Peer review of "Increase Paclitaxel Sensitivity to Better Suppress Serous Epithelial Ovarian Cancer via Ablating Androgen Receptor/Aryl Hydrocarbon Receptor-ABCG2 Axis"

_cancers, 2019, doi:10.3390/cancers11040463_

Round 1

Reviewer 1 Report

Drug resistance is the main problem in the treatment of cancers. Cancer cells can develop many different mechanisms protecting them against chemotherapy. Some of them are more universal, while others are present only in some drug resistant cells.

In the presents manuscript authors described a potential relation between expression of breast cancer resistant protein (BCRP) and resistance to paclitaxel, a drug used in the first line of ovarian cancer chemotherapy.

Authors performed a lot of different experiments to proof their hypothesis. However authors completely ignore one fact. According to literature data [A] and my own experience with drug resistant ovarian cancer cell lines [B, C ], the most important protein in resistance to paclitaxel is glycoprotein P (P-gp) encoded by MDR1 (ABCB1) gene. This fact should be described in Introduction and further discussed in Discussion section. However this is totally ignored by the Authors. In my opinion this is unacceptable.

I cannot agree with the sentence from introduction “The breast cancer resistance protein/ATP-binding cassette subfamily G member 2 (BCRP or ABCG2) belongs to the ABC transporter superfamily, which has been reported to be found in ovarian cancer stem cells  and is associated with paclitaxel-induced chemoresistance [12, 13]”, that BCRP is associated with paclitaxel resistance. In ref. 12 expression of BCRP was increased in CSCs and these cells were more resistant to paclitaxel, however they did not check expression of P-gp in investigated cell lines. Ref. 13 (my own) did not say that that BCRP expression is associated with PAC resistance.

I can also not agree with the sentence from results (lines 76, 77) “Given that previous studies have shown that paclitaxel sensitivity is influenced by ABCG2 expression and function [10, 27, 28],….”

and

from Discussion

“Sun et al. showed that AR and ABCG2 were upregulated under paclitaxel treatment and in 346 paclitaxel-resistant EOC cells [27, 28]. (lines 346-347)”.

Ref. 10 – describe a role of ABC transporters in steroid transport not in drug resistance.

Ref. 27 – is not about the role of ABCG2 in PAC resistance. In this paper ABCG2 in only one time present in all paper (in the table). In contrast ABCB1 is present in the text 11 times, and is described as a PAC resistant gene. In this paper expression of ABCB1 in PAC resistant cell line increased over 2500 fold and expression of ABCG2 increase only 18 fold. So the most important gen in PAC resistance described in this paper was ABCB1 not ABCG2.

In Ref. 28 impact of AR on expression of different PAC resistant genes is described. However in this paper silencing of AR have much more impact on ABCB1 gene then on ABCG2 gen further proofing that ABCB1 is much more important in PAC resistance than ABCG2.

In my opinion this is not a real scientific discussion but data manipulation by authors.

I would like the authors to compare the expression of MDR1/P-gp and BCRP at transcript and protein level in investigated cell lines. The same in OVCAR3 cell line resistant to PAC.

In all experiments expression of P-gp and BCRP should be investigated in parallel.

In a cytotoxicity study data should not be present as a absorbance but as a percent cell viability according to control designated as a 100%. This will be more clear.

In this type of experiments authors also showed that ASC-J9 increase PAC effectiveness bath in vitro and in vivo. However they do not explain mechanism of such results. According to authors hypothesis AR increase BCRP expression and ASC-J9 in AR degradation enhancer. So treatment with ASC-J9 should lead to BCRP downregulation. I would like to ask the authors to measure BCRP and MDR1/P-gp expression in OVCAR3 paclitaxel resistant cell line after ASC-J9 treatment at mRNA (q-PCR) and protein level (western blot) and in OVCAR3 paclitaxel resistant tumours by IHC.

The Discussion section is very poor and should be development.

It was very difficult to read this paper because of language quality, and all the text should be re-edited.

References:

A. Tomris Ozben. Mechanisms and strategies to overcome multiple drug resistance in cancer. FEBS Letters 580 (2006) 2903–2909.

B. Januchowski, R.; Sterzyńska, K.; Zaorska, K.; Sosińska, P.; Klejewski, A.; Brązert, M.; Nowicki, M.; Zabel, M. Analysis of MDR genes expression and cross-resistance in eight drug resistant ovarian cancer cell lines. J Ovarian Res. 2016, 9, 65. DOI: 10.1186/s13048-016-0278.

C. Januchowski, R.; Wojtowicz, K.; Sujka-Kordowska, P.; Andrzejewska, M.; Zabel, M. MDR gene expression analysis of six drug-resistant ovarian cancer cell lines. Biomed Res Int. 2013, 241763. DOI: 10.1155/2013/241763

Author Response

Response to reviewer 1:

Comment 1: Drug resistance is the main problem in the treatment of cancers. Cancer cells can develop many different mechanisms protecting them against chemotherapy. Some of them are more universal, while others are present only in some drug-resistant cells. In the presents manuscript authors described a potential relation between expression of breast cancer resistant protein (BCRP) and resistance to paclitaxel, a drug used in the first line of ovarian cancer chemotherapy. Authors performed a lot of different experiments to prove their hypothesis. However, the authors completely ignore one fact. According to literature data [A] and my own experience with drug-resistant ovarian cancer cell lines [B, C ], the most important protein in resistance to paclitaxel is glycoprotein P (P-gp) encoded by MDR1 (ABCB1) gene. This fact should be described in the Introduction and further discussed in the Discussion section. However, this is totally ignored by the Authors. In my opinion, this is unacceptable.

A. Tomris Ozben. Mechanisms and strategies to overcome multiple drug resistance in cancer. FEBS Letters 580 (2006) 2903–2909.

B. Januchowski, R.; Sterzyńska, K.; Zaorska, K.; Sosińska, P.; Klejewski, A.; Brązert, M.; Nowicki, M.; Zabel, M. Analysis of MDR genes expression and cross-resistance in eight drug-resistant ovarian cancer cell lines. , 9, 65. DOI: 10.1186/s13048-016-0278.

C. Januchowski, R.; Wojtowicz, K.; Sujka-Kordowska, P.; Andrzejewska, M.; Zabel, M. MDR gene expression analysis of six drug-resistant ovarian cancer cell lines. Biomed Res Int. 2013, 241763. DOI: 10.1155/2013/241763

Answer: We appreciate the reviewer's suggestions, and we recognize the reviewer’s effort to ABCB1 functions in ovarian cancer drug resistance. Furthermore, we also pay much attention to another ABC transporters such as ABCB1 and totally agree with literature data [A] and reviewer's experience with drug-resistant ovarian cancer cell lines [B, C]. Those studies [A, B, C] are valuable and provided a high impact on ovarian cancer drug resistance. However, the authors used A2780 cells as an in vitro model in reference [B] (derived from ovarian endometrioid adenocarcinoma, Ref: European Collection of Authenticated Cell Cultures (ECACC), https://www.phe-culturecollections.org.uk/media/113526/a2780-cell-line-profile.pdf) and primary ovarian cancer cell line in reference [C] which did not further identify the subtype of ovarian cancer. And also, these two studies did not check AR expression in the cell lines they used. These studies indeed provided us with a highly valuable reference for drug resistance field and our research. However, in our manuscripts, we emphasized alternative AR transactivation role in AR-positive EOC serous subtype. And base on our previous studies in ovarian teratoma cancer stem/progenitor cells [1, 2], we are more interested in ABCG2 (one of potential ovarian cancer stem cell markers) expression which involving in AR transactivation regulation. Although our study focuses on AR-ABCG2-chemoresistance, we totally agree with the reviewer's opinion.

Comment 2: I cannot agree with the sentence from introduction “The breast cancer resistance protein/ATP-binding cassette subfamily G member 2 (BCRP or ABCG2) belongs to the ABC transporter superfamily, which has been reported to be found in ovarian cancer stem cells  and is associated with paclitaxel-induced chemoresistance [12, 13]", that BCRP is associated with paclitaxel resistance. In ref. 12 expression of BCRP was increased in CSCs and these cells were more resistant to paclitaxel, however, they did not check the expression of P-gp in investigated cell lines. Ref. 13 (my own) did not say that that BCRP expression is associated with PAC resistance.

Answer: We apologized for careless handling of the sentence from the introduction. The sentence and citation are revised as follow

The breast cancer resistance protein/ATP-binding cassette subfamily G member 2 (BCRP or ABCG2) belongs to the ABC transporter superfamily, which has been reported to be found in ovarian cancer stem cells and is associated with chemoresistance [12, 13]”.

Comment 3: I can also not agree with the sentence from results (lines 76, 77) “Given that previous studies have shown that paclitaxel sensitivity is influenced by ABCG2 expression and function [10, 27, 28],….” And from Discussion “Sun et al. showed that AR and ABCG2 were upregulated under paclitaxel treatment and in 346 paclitaxel-resistant EOC cells [27, 28]. (lines 346-347)”. Ref. 10 – describe the role of ABC transporters in steroid transport not in drug resistance. Ref. 27 – is not about the role of ABCG2 in PAC resistance. In this paper ABCG2 in only one time present in all paper (in the table). In contrast, ABCB1 is present in the text 11 times, and is described as a PAC resistant gene. In this paper expression of ABCB1 in PAC resistant cell line increased over 2500 fold and expression of ABCG2 increase only 18 fold. So the most important gen in PAC resistance described in this paper was ABCB1, not ABCG2.  In Ref. 28 impact of AR on the expression of different PAC resistant genes is described. However, in this paper silencing of AR have much more impact on ABCB1 gene then on ABCG2 gen further proofing that ABCB1 is much more important in PAC resistance than ABCG2.

Answer: We apologized for careless handling of sentence and citation. The mistakes are revised as follow

Given that previous studies have shown that paclitaxel treatment changed ABCG2 expression and function [26, 27].

Besides, we also revised the discussion section as follow.

“Except for ABCG2, there is another ABC transporters associated with resistance of chemo-reagents [3]. Especially, ABCB1 had been reported associated with paclitaxel resistance in ovarian cancer [4-6]. Januchowski et al established eight drug resistant cell lines from A2780 ovarian cancer cell line those were resistant to cisplatin, paclitaxel and doxorubicin. And the authors demonstrated that ABCB1 expression was upregulated in paclitaxel-resistant A2780 cells [5]. Furthermore, the same research group presented consistent results in six drug-resistant ovarian cancer cell lines derived from primary culture [6]. Together, the authors concluded that ABCB1 plays a critical role in paclitaxel resistance [5, 6]. On the other hand, Sun et al determined ABCB1 gene expression and regulation in paclitaxel-resistant ovarian cancer (SKOV3) [7]. They found ABCB1 expression was much higher in resistant cells than parental cells and AR as well. In this study, authors identified that both DHT and paclitaxel activated AR transactivity and regulated ABCB1 gene expression. To further evaluated AR directly promoted ABCB1 expression, the ABCB1 promoter region was analyzed. They identified several classical AREs existed in promoter region and showed directly regulation of AR for ABCB1 expression [7]. In these studies, the results indicated that ABCB1 played a major role in paclitaxel resistance [5, 6] and AR was also involved in this axis [7]. However, besides ABCB1, another minor potential ABC transporters could also involve in paclitaxel resistance. Sun et al showed that ABCG2 expression was higher in resistance cells than parental cells by using transcriptome analysis. And the transcriptome data also presented that ABCG2 associated with AR expression [8, 9].     

In previous studies, Bailey et al analyzed ABCG2 promoter region and data showed none of classical ARE existed in the promoter [10]. Consistently, Scotto KW reviewed transcription factors of ABC transporters [11]. This report presented all potential transcription factors for every ABC transporters including ABCB1 and ABCG2. However, AR was not mentioned in transcription factors analysis of all ABC transporters [11]. According to accumulated studies, AR was showed its important role in regulating ABCB1 expression [7-9]. In our study, we performed different AR transactiviation function in regulating ABCG2 expression. Even though ABCG2 plays a minor role in paclitaxel resistance, our study showed the novel transactivation actions in ABCG2 expression including paclitaxel-induced transactivity, alternative ARE existed in ABCG2 promoter region and recruitment of AhR in AR-ABCG2 axis.  Base on previous studies and our findings, these data implied another effective therapeutic possibility of targeting AR in treating ovarian cancer.

Comment 4: In my opinion, this is not a real scientific discussion but data manipulation by authors.

I would like the authors to compare the expression of MDR1/P-gp and BCRP at transcript and protein level in investigated cell lines. The same in OVCAR3 cell line resistant to PAC. In all experiments expression of P-gp and BCRP should be investigated in parallel.

Answer: Thanks to reviewer’s good suggestion. The question of MDR1/P-gp resistant to PAC had been reported by Sun et al [12]. However, we performed to analyze the alternative AR transactivity role in regulation ABCG2 and degradation of AR is an effective therapeutic strategy for AR-positive ovarian cancer serous subtype.

Comment 5: In a cytotoxicity study data should not be present as an absorbance but as a percent cell viability according to control designated as a 100%. This will be more clear.

Answer: Thanks to reviewer’s suggestion. The figure presentation of cytotoxicity study is revised according to the reviewer's suggestion and as shown below, the percent cell viability according to control designated as a 100%.

Comment 6: In this type of experiments authors also showed that ASC-J9 increase PAC effectiveness bath in vitro and in vivo. However, they do not explain the mechanism of such results. According to the authors' hypothesis, AR increases BCRP expression and ASC-J9 in AR degradation enhancer. So treatment with ASC-J9 should lead to BCRP downregulation. I would like to ask the authors to measure BCRP and MDR1/P-gp expression in OVCAR3 paclitaxel-resistant cell line after ASC-J9 treatment at mRNA (q-PCR) and protein level (western blot) and in OVCAR3 paclitaxel-resistant tumors by IHC. 

Answer: Thanks to reviewer’s suggestion. Since MDR1/P-gp had been reported by Sun et al., here, we follow the reviewer's suggestion that measures BCRP mRNA and protein expression after ASC-J9 treatment. And data showed as below. Left panel showed BCRP mRNA expression under ASC-J9 treatment in paclitaxel-resistant OVCAR3 cells. The data indicated that BCRP mRNA expression was decreased by treating ASC-J9 compared to DMSO control. Right panel showed BCRP protein expression under ASC-J9 treatment in paclitaxel-resistant cells. And OVCAR3 AR overexpression cells were used as positive control. In this data, we confirmed that BCRP protein expression was suppressed under treatment of ASC-J9. Both mRNA and protein expression of BCRP showed consistence and demonstrated that treatment of ASC-J9 leaded BCRP downregulation. 

Comment 7: The Discussion section is very poor and should be developed. It was very difficult to read this paper because of language quality, and all the text should be re-edited.

Answer: Thanks to reviewer’s suggestion. We have revised the discussion section with professional English editing in the re-submitting manuscript.

References:

1.         Chung, W.M., et al., Ligand-independent androgen receptors promote ovarian teratocarcinoma cell growth by stimulating self-renewal of cancer stem/progenitor cells. Stem Cell Res, 2014. 13(1): p. 24-35.

2.         Chung, W.M., et al., MicroRNA-21 promotes the ovarian teratocarcinoma PA1 cell line by sustaining cancer stem/progenitor populations in vitro. Stem Cell Res Ther, 2013. 4(4): p. 88.

3.         Wilkens, S., Structure and mechanism of ABC transporters. F1000Prime Rep, 2015. 7: p. 14.

4.         Ozben, T., Mechanisms and strategies to overcome multiple drug resistance in cancer. FEBS Lett, 2006. 580(12): p. 2903-9.

5.         Januchowski, R., et al., Analysis of MDR genes expression and cross-resistance in eight drug resistant ovarian cancer cell lines. J Ovarian Res, 2016. 9(1): p. 65.

6.         Januchowski, R., et al., MDR gene expression analysis of six drug-resistant ovarian cancer cell lines. Biomed Res Int, 2013. 2013: p. 241763.

7.         Sun, N.K., et al., Androgen receptor transcriptional activity and chromatin modifications on the ABCB1/MDR gene are critical for taxol resistance in ovarian cancer cells. J Cell Physiol, 2019. 234(6): p. 8760-8775.

8.         Sun, N.K., et al., Transcriptomic profiling of taxol-resistant ovarian cancer cells identifies FKBP5 and the androgen receptor as critical markers of chemotherapeutic response. Oncotarget, 2014. 5(23): p. 11939-56.

9.         Sun, N.K., et al., Integrative transcriptomics-based identification of cryptic drivers of taxol-resistance genes in ovarian carcinoma cells: Analysis of the androgen receptor. Oncotarget, 2015. 6(29): p. 27065-82.

10.       Bailey-Dell, K.J., et al., Promoter characterization and genomic organization of the human breast cancer resistance protein (ATP-binding cassette transporter G2) gene. Biochim Biophys Acta, 2001. 1520(3): p. 234-41.

11.       Scotto, K.W., Transcriptional regulation of ABC drug transporters. Oncogene, 2003. 22(47): p. 7496-511.

12.       Sun, N.K., et al., Androgen receptor transcriptional activity and chromatin modifications on the ABCB1/MDR gene are critical for taxol resistance in ovarian cancer cells. J Cell Physiol, 2018.

Reviewer 2 Report

In the manuscript by Chung et al., Authors have tried to elucidate the involvement of androgen receptor (AR) in the mechanism of resistance to paclitaxel in epithelial ovarian cancer (EOC) cells by means of some methods, such as co-IP, luciferase reporter assay and ChIP assay in vitro, and IHC staining by AR antibody in vivo (patients). They also used AR degradation enhancer (ASC-J9) to examine paclitaxel-associated and paclitaxel-resistant cytotoxicity in vitro and in vivo. The authors state that the results obtained demonstrated that activation of AR transactivity beyond the androgen-associated biological effect and conclude that the degradation of AR could be a most effective therapeutic strategy for treating AR-positive EOC serous subtype.

The study appears well conducted and the results obtained by effective and convincing methods are consistent with the hypothesis of the rationale.

Authors made many affords to demonstrate this mechanism of resistance to paclitaxel.

However, I have some concerns.

In Fig.2B and 2C are reported the reduced cytotoxicity by increasing AR expression via adding AR-cDNA (Figure 2B), and the increased cytotoxicity by decreasing AR expression via adding AR-shRNA (Figure 2C).

However, from these plots is difficult to understand a significant difference with the vector control in all three cell lines. Even the IC50 values of these treatments of each manipulated cells was showed in supplementary Table S4 without statistical evaluations.

Therefore, I suggest to improve the presentation of these Figures 2B and 2C, by changing the scale of axis of drug concentration. For example, to show data up to 0.1 µM or at lower concentrations, since the greatest effects are obtained in the low nanomolar range, as confirmed by the IC50 values of supplementary Table S4, that should in turn be implemented with the analysis of the P value.

The same considerations could be done for the data reported in Fig. 5A and 5C.

Fig.4, panels of Fig D. It is not clear to what blots are related the lower panels of the quantitation results of ChIP experiments. Please explain better.

Many acronysms in the abstract, and some of them are not as well known and familiar to many readers.

Some editorial oversights; e.g., line 30, tansactivity instead of transactivity; line 34 explain instead of explains; etc..

Author Response

Response to reviewer 2:

In the manuscript by Chung et al., Authors have tried to elucidate the involvement of androgen receptor (AR) in the mechanism of resistance to paclitaxel in epithelial ovarian cancer (EOC) cells by means of some methods, such as co-IP, luciferase reporter assay and ChIP assay in vitro, and IHC staining by AR antibody in vivo (patients). They also used AR degradation enhancer (ASC-J9) to examine paclitaxel-associated and paclitaxel-resistant cytotoxicity in vitro and in vivo. The authors state that the results obtained demonstrated that activation of AR transactivity beyond the androgen-associated biological effect and conclude that the degradation of AR could be the most effective therapeutic strategy for treating AR-positive EOC serous subtype. The study appears well conducted and the results obtained by effective and convincing methods are consistent with the hypothesis of the rationale. Authors made many afford to demonstrate this mechanism of resistance to paclitaxel. However, I have some concerns.

Comment 1: In Fig.2B and 2C are reported the reduced cytotoxicity by increasing AR expression via adding AR-cDNA (Figure 2B), and the increased cytotoxicity by decreasing AR expression via adding AR-shRNA (Figure 2C).However, from these plots is difficult to understand a significant difference with the vector control in all three cell lines. Even the IC50 values of these treatments of each manipulated cells were shown in Supplementary Table S4 without statistical evaluations. Therefore, I suggest improving the presentation of these Figures 2B and 2C, by changing the scale of the axis of drug concentration. For example, to show data up to 0.1 µM or at lower concentrations, since the greatest effects are obtained in the low nanomolar range, as confirmed by the IC50 values of supplementary Table S4, that should, in turn, be implemented with the analysis of the P value.  The same considerations could be done for the data reported in Fig. 5A and 5C.

Answer: Thanks to reviewer’s good suggestion. However, we try to describe that AR effect on cell cytotoxicity not only in low concentration of paclitaxel but also in high concentration. At high concentration (0.1~1 uM) of paclitaxel treatment, overexpression of AR showed more resistance than vector control (fig 2B). And we also try to show ASC-J9 effects at higher concentration of treatment in figure 5A and C. Here, we referred to the reviewer's suggestion, but we modify the Y axis as survival (100% control) in figure 2B, 2C, and figure 5A, 5C that show much significance difference of cytotoxicity from control and AR-manipulated cell lines. Supplementary Table S4 had been revised according to reviewer’s suggestion. And modified figures and table are showed as below and presented in the manuscript.

Comment 2: Fig.4, panels of Fig D. It is not clear to what blots are related to the lower panels of the quantitation results of ChIP experiments. Please explain better.

Answer: Thanks to reviewer’s good suggestion. We follow the reviewer's suggestion that show a clear indication for quantitation results of ChIP experiments in figure 4D. The revised figures are showed as below and presented in the manuscript.

Comment 3: Many acronyms in the abstract, and some of them are not as well known and familiar to many readers.

Answer: Thanks to reviewer’s good suggestion. We revised acronyms in the abstract that is clear for reading.

Some editorial oversights; e.g., line 30, transactivity instead of transactivity; line 34 explain instead of explains; etc.

Answer: We apologized for careless handling of manuscript editing. We corrected wrongly written in manuscript more carefully and thank you for the reviewer's revising.

Reviewer 3 Report

The article “Increase Paclitaxel Sensitivity to Better Suppress Serous Epithelial Ovarian Cancer via Ablating Androgen Receptor/Aryl Hydrocarbon Receptor ABCG2 Axis” by Chung et al demonstrated the functional role of AR/AhR ABCG2 axis in paclitaxel resistance, as demonstrated by in vitro AR over expression / knock down experiments in cell lines. They have also demonstrated that blocking AR/AhR interaction can sensitize cells to paclitaxel therapy. The study is interesting, and most of the results were supported by experimental data. Reviewer feels the manuscript may be published after minor revision (listed below).

The authors showed AR protein expression was predominantly in Serous sub type of ovarian cancer, compared to other histotypes, (Figure 3) and there was a correlation for expression of AR and ABCG2 in serous ovarian cancer. Did the authors evaluate the expression of AR in cell lines of other ovarian cancer histotypes (especially clear cell ovarian cancer)? Would you think AR/AhR ABCG2 axis is important only in serous sub type? Does AR/AhR ABCG2 axis plays any role in resistance against platinum-based therapies? Within Serous sub-type, is there any expression difference for AR in resistance tumor versus refractory cases. Reviewer feels the authors should discuss these aspects in detail. 

Author Response

Response to reviewer 3:

The article “Increase Paclitaxel Sensitivity to Better Suppress Serous Epithelial Ovarian Cancer via Ablating Androgen Receptor/Aryl Hydrocarbon Receptor ABCG2 Axis” by Chung et al demonstrated the functional role of AR/AhR ABCG2 axis in paclitaxel resistance, as demonstrated by in vitro AR overexpression/knockdown experiments in cell lines. They have also demonstrated that blocking AR/AhR interaction can sensitize cells to paclitaxel therapy. The study is interesting, and most of the results were supported by experimental data. The reviewer feels the manuscript may be published after minor revision (listed below).

The authors showed AR protein expression was predominantly in Serous subtype of ovarian cancer, compared to other histotypes, (Figure 3) and there was a correlation for expression of AR and ABCG2 in serous ovarian cancer.

Comment 1: Did the authors evaluate the expression of AR in cell lines of other ovarian cancer histotypes (especially clear cell ovarian cancer)?

Answer: Thanks to reviewer’s good question. Actually, excepting three cell lines, we also evaluate AR expression in other serous subtype cell lines such as SKOV3, OVCAR429 (data not shown). However, to answer reviewer’s question, Sun et al showed AR expression in serous (SKOV3), clear cell (TOV-21G) and endometrioid (MDAH2774) subtype cells and all of the cells presented AR expression8.

Comment 2: Would you think AR/AhR ABCG2 axis is important only in serous subtype?

Answer: Thanks to reviewer’s good question. We found the AR/AhR-ABCG2 axis is important in EOC serous subtype. However, whether AR/AhR-ABCG2 axis is critical in other cancer had been implying in other studies. Previous studies had shown that AhR is one of the transcriptional activators of human ABCG2 expression9 and AhR activity associated with ABCG2 expression and function in many cancer cells such as liver cancer cells7, breast cancer cells6, colon cancer cells10. In castration-resistant prostate cancer, studies demonstrated that AhR participates in crosstalk with the AR signaling5, 11. According to these studies, AR/AhR-ABCG2 axis showed its significance not only in EOC serous subtype but also other cancers.

Comment 3: Does AR/AhR ABCG2 axis plays any role in resistance against platinum-based therapies?

Answer: Thanks to reviewer’s good question. In our findings, paclitaxel is not only a chemoreagent but also an AR inducer for developing serous subtype chemoresistance. Therefore, the AR/AhR-ABCG2 axis plays an important role in paclitaxel-induced chemoresistance. In our data, we are not sure whether AR/AhR-ABCG2 plays any role in resistance against platinum-based therapies. However, there are some studies showed that ABCG2 expression associated with cisplatin treatment and cytotoxicity in cancers2, 4, including ovarian cancer12, 13. Especially, Hsieh et al showed that overexpression of AR upregulated ABCG2 expression and reduce the cytotoxic effect of chemotherapeutic drugs, including cisplatin in upper urinary tract urothelial carcinomas (UUTUCs)2. Although Hsieh’s group presented AR-ABCG2 against platinum-based therapy in UUTUCs, the results somehow can imply that targeting on AR could be a potential therapeutic strategy for AR/AhR ABCG2 axis-associated platinum resistance.

Comment 4: Within Serous sub-type, is there any expression difference for AR in resistance tumor versus refractory cases. The reviewer feels the authors should discuss these aspects in detail.

Answer: Thanks to reviewer’s good question. We searched for any possible publications associated with AR expression in resistant and refractory cases. However, there is no related report that compared resistant and refractory patients at the same conditions. Here, we only found that Feng et al reported hormone receptor expression pattern in primary and recurrent high-grade ovarian cancer1. In this study, retrospectively examined 107 high-grade serous ovarian cancer (HGSC) patients with paired primary and recurrent tumor specimens. Patient characteristics described as follow, 107 HGSC patients had a median (range) follow-up time of 42 (4–115) months. 103 (96.3%) patients had advanced stage disease. At the time of primary surgery, 30 (28%) were debulked to no macroscopic residual disease, and 57 (53.3%) were debulked to <1 cm of macroscopic disease. 106 patients received platinum-based adjuvant chemotherapy, and all patients underwent secondary debulking surgery for ovarian cancer recurrence with a median (range) PFS of 15 (5–66) months. At the time of secondary cytoreduction, 60 (56.1%) were debulked to no macroscopic residual disease, and 25 (23.4%) were debulked to <1 cm of macroscopic disease. Surprisingly, the authors examined the expression of hormone receptors by immunohistochemistry (n=107) and found the Androgen receptor (AR) was downregulated from 33.6% (primary) to 17.5% (recurrent), respectively. On the other hand, Kashiwagi et al investigated AR expression by immunohistochemical stain in tissue microarrays (TMAs) consisting of muscle-invasive bladder cancer specimens from patients who subsequently received neoadjuvant GC therapy3. Authors compared their expression levels between responders versus non-responders to chemotherapy and the results showed that overall, AR was positive in 19 (35%) of 55 cases, including 5 (21%) of 24 responders and 14 (45%) of 31 non-responders. Thus, the author demonstrated that AR positivity tended to correlate with resistance to chemotherapy (P = 0.087)3. Although AR expression in resistant tumor and the refractory tumor was reported in two different cancer, we considered that AR expression tent to chemoresistance than recurrence cases.

References:

1          Feng Z, Wen H, Ju X, Bi R, Chen X, Yang W et al. Hormone receptor expression profiles differ between primary and recurrent high-grade serous ovarian cancers. Oncotarget 2017; 8: 32848-32855.

2          Hsieh TF, Chen CC, Yu AL, Ma WL, Zhang C, Shyr CR et al. Androgen receptor decreases the cytotoxic effects of chemotherapeutic drugs in upper urinary tract urothelial carcinoma cells. Oncology letters 2013; 5: 1325-1330.

3          Kashiwagi E, Ide H, Inoue S, Kawahara T, Zheng Y, Reis LO et al. Androgen receptor activity modulates responses to cisplatin treatment in bladder cancer. Oncotarget 2016; 7: 49169-49179.

4          Li X, Zou Z, Tang J, Zheng Y, Liu Y, Luo Y et al. NOS1 upregulates ABCG2 expression contributing to DDP chemoresistance in ovarian cancer cells. Oncology letters 2019; 17: 1595-1602.

5          Ohtake F, Fujii-Kuriyama Y, Kato S. AhR acts as an E3 ubiquitin ligase to modulate steroid receptor functions. Biochem Pharmacol 2009; 77: 474-484.

6          Salisbury TB, Tomblin JK, Primerano DA, Boskovic G, Fan J, Mehmi I et al. Endogenous aryl hydrocarbon receptor promotes basal and inducible expression of tumor necrosis factor target genes in MCF-7 cancer cells. Biochem Pharmacol 2014; 91: 390-399.

7          Satsu H, Yoshida K, Mikubo A, Ogiwara H, Inakuma T, Shimizu M. Establishment of a stable aryl hydrocarbon receptor-responsive HepG2 cell line. Cytotechnology 2015; 67: 621-632.

8          Sun NK, Kohli A, Huang SL, Chang TC, Chao CC. Androgen receptor transcriptional activity and chromatin modifications on the ABCB1/MDR gene are critical for taxol resistance in ovarian cancer cells. Journal of cellular physiology 2018.

9          Tan KP, Wang B, Yang M, Boutros PC, Macaulay J, Xu H et al. Aryl hydrocarbon receptor is a transcriptional activator of the human breast cancer resistance protein (BCRP/ABCG2). Mol Pharmacol 2010; 78: 175-185.

10        Tompkins LM, Li H, Li L, Lynch C, Xie Y, Nakanishi T et al. A novel xenobiotic responsive element regulated by aryl hydrocarbon receptor is involved in the induction of BCRP/ABCG2 in LS174T cells. Biochem Pharmacol 2010; 80: 1754-1761.

11        Tran C, Richmond O, Aaron L, Powell JB. Inhibition of constitutive aryl hydrocarbon receptor (AhR) signaling attenuates androgen independent signaling and growth in (C4-2) prostate cancer cells. Biochem Pharmacol 2013; 85: 753-762.

12        Xiong X, Arvizo RR, Saha S, Robertson DJ, McMeekin S, Bhattacharya R et al. Sensitization of ovarian cancer cells to cisplatin by gold nanoparticles. Oncotarget 2014; 5: 6453-6465.

13        Zhang W, Yu F, Wang Y, Zhang Y, Meng L, Chi Y. Rab23 promotes the cisplatin resistance of ovarian cancer via the Shh-Gli-ABCG2 signaling pathway. Oncology letters 2018; 15: 5155-5160.

Round 2

Reviewer 1 Report

Authors improved the quality of manuscript especially a discussion section. In my opinion  comparision of ABCB1 and ABCG2 espression could improve the quality of research, however author give explanation in Discussion section.